# EFFICIENT ARCHITECTURE SEARCH FOR CONTINUAL LEARNING

## ABSTRACT

Continual learning with neural networks is an important learning framework in AI that aims to learn a sequence of tasks well. However, it is often confronted with three challenges: (1) overcome the catastrophic forgetting problem, (2) adapt the current network to new tasks, and meanwhile (3) control its model complexity. To reach these goals, we propose a novel approach named as Continual Learning with Efficient Architecture Search, or CLEAS in short. CLEAS works closely with neural architecture search (NAS) which leverages reinforcement learning techniques to search for the best neural architecture that fits a new task. In particular, we design a *neuron-level* NAS controller that decides which old neurons from previous tasks should be reused (knowledge transfer), and which new neurons should be added (to learn new knowledge). Such a fine-grained controller allows finding a very concise architecture that can fit each new task well. Meanwhile, since we do not alter the weights of the reused neurons, we perfectly memorize the knowledge learned from previous tasks. We evaluate CLEAS on numerous sequential classification tasks, and the results demonstrate that CLEAS outperforms other state-of-the-art alternative methods, achieving higher classification accuracy while using simpler neural architectures.

## 1 INTRODUCTION

Continual learning, or lifelong learning, refers to the ability of continually learning new tasks and also performing well on learned tasks. It has attracted enormous attention in AI as it mimics a human learning process - constantly acquiring and accumulating knowledge throughout their lifetime (Parisi et al., 2019). Continual learning often works with deep neural networks (Javed & White, 2019; Nguyen et al., 2017; Xu & Zhu, 2018) as the flexibility in a network design can effectively allow knowledge transfer and knowledge acquisition. However, continual learning with neural networks usually faces three challenges. The first one is to overcome the so-called *catastrophic forgetting* problem (Kirkpatrick et al., 2017), which states that the network may forget what has been learned on previous tasks. The second one is to effectively adapt the current network parameters or architecture to fit a new task, and the last one is to control the network size so as not to generate an overly complex network.

In continual learning, there are two main categories of strategies that attempt to solve the aforementioned challenges. The first category is to train all tasks within a network with fixed capacity. For example, (Rebuffi et al., 2017; Lopez-Paz & Ranzato, 2017; Aljundi et al., 2018) replay some old samples with the new task samples and then learn a new network from the combined training set. The drawback is that they typically require a memory system that stores past data. (Kirkpatrick et al., 2017; Liu et al., 2018) employ some regularization terms to prevent the re-optimized parameters from deviating too much from the previous ones. Approaches using fixed network architecture, however, cannot avoid a fundamental dilemma - they must either choose to retain good model performances on learned tasks, leaving little room for learning new tasks, or compromise the learned model performances to allow learning new tasks better.

To overcome such a dilemma, the second category is to expand the neural networks dynamically (Rusu et al., 2016; Yoon et al., 2018; Xu & Zhu, 2018). They typically fix the parameters of the old neurons (partially or fully) in order to eliminate the forgetting problem, and also permit adding new neurons to adapt to the learning of a new task. In general, expandable networks can achieve better model performances on all tasks than the non-expandable ones. However, a new issue appears: expandable

networks can gradually become overly large or complex, which may break the limits of the available computing resources and/or lead to over-fitting.

In this paper, we aim to solve the continual learning problems by proposing a new approach that only requires *minimal* expansion of a network so as to achieve high model performances on both learned tasks and the new task. At the heart of our approach we leverage Neural Architecture Search (NAS) to find a very concise architecture to fit each new task. Most notably, we design NAS to provide a *neuron-level* control. That is, NAS selects two types of *individual* neurons to compose a new architecture: (1) a subset of the previous neurons that are most useful to modeling the new task; and (2) a minimal number of new neurons that should be added. Reusing part of the previous neurons allows efficient knowledge transfer; and adding new neurons provides additional room for learning new knowledge. Our approach is named as Continual Learning with Efficient Architecture Search, or CLEAS in short. Below are the main features and contributions of CLEAS.

- CLEAS dynamically expands the network to adapt to the learning of new tasks and uses NAS to determine the new network architecture;

- CLEAS achieves zero forgetting of the learned knowledge by keeping the parameters of the previous architecture unchanged;

- NAS used in CLEAS is able to provide a neuron-level control which expands the network minimally. This leads to an effective control of network complexity;

- The RNN-based controller behind CLEAS is using an entire network configuration (with all neurons) as a state. This state definition deviates from the current practice in related problems that would define a state as an observation of a single neuron. Our state definition leads to improvements of 0.31%, 0.29% and 0.75% on three benchmark datasets.

- If the network is a convolutional network (CNN), CLEAS can even decide the best filter size that should be used in modeling the new task. The optimized filter size can further improve the model performance.

We start the rest of the paper by first reviewing the related work in Section 2. Then we detail our CLEAS design in Section 3. Experimental evaluations and the results are presented in Section 4.

## 2 RELATED WORK

**Continual Learning** Continual learning is often considered as an online learning paradigm where new skills or knowledge are constantly acquired and accumulated. Recently, there are remarkable advances made in many applications based on continual learning: sequential task processing (Thrun, 1995), streaming data processing (Aljundi et al., 2019), self-management of resources (Parisi et al., 2019; Diethe et al., 2019), etc. A primary obstacle in continual learning, however, is the *catastrophic forgetting* problem and many previous works have attempted to alleviate it. We divide them into two categories depending on whether their networks are expandable.

The first category uses a large network with fixed capacity. These methods try to retain the learned knowledge by either replaying old samples (Rebuffi et al., 2017; Rolnick et al., 2019; Robins, 1995) or enforcing the learning with regularization terms (Kirkpatrick et al., 2017; Lopez-Paz & Ranzato, 2017; Liu et al., 2018; Zhang et al., 2020). Sample replaying typically requires a memory system which stores old data. When learning a new task, part of the old samples are selected and added to the training data. As for regularized learning, a representative approach is Elastic Weight Consolidation (EWC) (Kirkpatrick et al., 2017) which uses the Fisher information matrix to regularize the optimization parameters so that the important weights for previous tasks are not altered too much. Other methods like (Lopez-Paz & Ranzato, 2017; Liu et al., 2018; Zhang et al., 2020) also address the optimization direction of weights to prevent the network from forgetting the previously learned knowledge. The major limitation of using fixed networks is that it cannot properly balance the learned tasks and new tasks, resulting in either forgetting old knowledge or acquiring limited new knowledge.

To address the above issue, another stream of works propose to dynamically expand the network, providing more room for obtaining new knowledge. For example, Progressive Neural Network (PGN) (Rusu et al., 2016) allocates a fixed number of neurons and layers to the current model for a new task. Apparently, PGN may end up generating an overly complex network that has high redundancy and it can easily crash the underlying computing system that has only limited resources.

Another approach DEN (Dynamically Expandable Network) (Yoon et al., 2018) partially mitigates the issue of PGN by using group sparsity regularization techniques. It strategically selects some old neurons to retrain, and adds new neurons only when necessary. However, DEN can have the forgetting problem due to the retraining of old neurons. Another drawback is that DEN has very sensitive hyperparameters that need sophisticated tuning. Both of these algorithms only grow the network and do not have a neuron level control which is a significant departure from our work. Most recently, a novel method RCL (Reinforced Continual Learning) (Xu & Zhu, 2018) also employs NAS to expand the network and it can further decrease model complexity. The main difference between RCL and CLEAS is that RCL blindly reuses all the neurons from all of the previous tasks and only uses NAS to decide how many new neurons should be added. However, reusing all the old neurons has two problems. First, it creates a lot of redundancy in the new network and some old neurons may even be misleading and adversarial; second, excessively many old neurons reused in the new network can dominate its architecture, which may significantly limit the learning ability of the new network. Therefore, RCL does not really optimize the network architecture, thus it is unable to generate an efficient and effective network for learning a new task. By comparison, CLEAS designs a fine-grained NAS which provides *neuron-level* control. It optimizes every new architecture by determining whether to reuse each old neuron and how many new neurons should be added to each layer.

**Neural Architecture Search**   NAS is another promising research topic in the AI community. It employs reinforcement learning techniques to automatically search for a desired network architecture for modeling a specific task. For instance, Cai et al. (Cai et al., 2018) propose EAS to discover a superb architecture with a reinforced meta-controller that can grow the depth or width of a network; Zoph et al. (Zoph & Le, 2016) propose an RNN-based controller to generate the description of a network, and the controller is reinforced by the predicting accuracy of a candidate architecture. Pham et al. (Pham et al., 2018) propose an extension of NAS, namely ENAS, to speed up training processing by forcing all child networks to share weights. Apart from algorithms, NAS also has many valuable applications such as image classification (Real et al., 2019; Radosavovic et al., 2019), video segmentation (Nekrasov et al., 2020), text representation (Wang et al., 2019) and etc. Hence, NAS is a demonstrated powerful tool and it is especially useful in continual learning scenarios when one needs to determine what is a good architecture for the new task.

## 3 METHODOLOGY

There are two components in the CLEAS framework: one is the *task network* that continually learns a sequence of tasks; and the other is *controller network* that dynamically expands the task network. The two components interact with each other under the reinforcement learning context - the task network sends the controller a reward signal which reflects the performance of the current architecture design; the controller updates its policy according to the reward, and then generates a new architecture for the task network to test its performance. Such interactions repeat until a good architecture is found. Figure 1 illustrates the overall structure of CLEAS. On the left is the task network, depicting an optimized architecture for task $t-1$ (it is using gray and pink neurons) and a candidate architecture for task $t$. They share the same input neurons but use their own output neurons. Red circles are newly added neurons and pink ones are reused neurons from task $t-1$ (or any previous task). To train the network, only the red weights that connect new-old or new-new neurons are optimized. On the right is the controller network which implements an RNN. It provides a neuron-level control to generate a description of the task network design. Each blue square is an RNN cell that decides to use or drop a certain neuron in the task network.

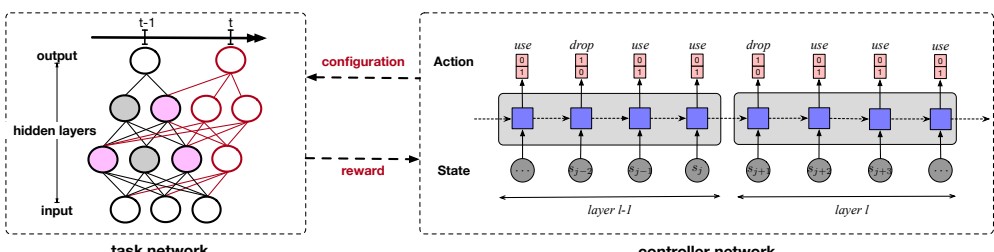

Figure 1: The framework of CLEAS.

### 3.1 NEURAL ARCHITECTURE SEARCH MODEL

**Task Network**  The task network can be any neural network with expandable ability, for example, a fully connected network or a CNN, etc. We use the task network to model a sequence of tasks. Formally, suppose there are $T$ tasks and each has a training set $\mathcal{D}_t = \{(x_i, y_i)\}_{i=1}^{O_t}$, a validation set $\mathcal{V}_t = \{(x_i, y_i)\}_{i=1}^{M_t}$ and a test set $\mathcal{T}_t = \{(x_i, y_i)\}_{i=1}^{K_t}$, for $t = 1, 2, \ldots, T$. We denote by $A_t$ the network architecture that is generated to model task $t$. Moreover, we denote $A_t = (N_t, W_t)$ where $N_t$ are the neurons or filters used in the network and $W_t$ are the corresponding weights. We train the first task with a basic network $A_1$ by solving the standard supervised learning problem

$$\overline{W}_1 = \arg\min_{W_1} \mathbb{L}_1(W_1; \mathcal{D}_1), \tag{1}$$

where $\mathbb{L}_1$ is the loss function for the first task. For the optimization procedure, we use stochastic gradient descent (SGD) with a constant learning rate. The network is trained till the required number of epochs or convergence is reached.

When task $t$ ($t > 1$) arrives, for every task $k < t$ we already know its optimized architecture $A_k$ and parameters $\overline{W}_k$. Now we use the controller to decide a network architecture for task $t$. Consider a candidate network $A_t = (N_t, W_t)$. There are two types of neurons in $N_t$: $N_t^{old}$ are used neurons from previous tasks and $N_t^{new} = N_t \setminus N_t^{old}$ are the new neurons added. Based on this partition, the weights $W_t$ can be also divided into two disjoint sets: $W_t^{old}$ are the weights that connect only used neurons, and $W_t^{new} = W_t \setminus W_t^{old}$ are the new weights that connect old-new or new-new neurons. Formally, $N_t^{old} = \{n \in N_t \mid \text{ there exists } k < t \text{ such that } n \in N_k\}$ and $W_t^{old} = \{w \in W_t \mid \text{ there exists } n_1, n_2 \in N_t^{old} \text{ such that } w \text{ connects } n_1, n_2\}$. The training procedure for the new task is to only optimize the new weights $W_t^{new}$ and leave $W_t^{old}$ unchanged, equal to their previously optimized values $\overline{W}_t^{old}$. Therefore, the optimization problem for the new task reads

$$\overline{W}_t^{new} = \arg\min_{W_t^{new}} \mathbb{L}_t(W_t|_{W_t^{old} = \overline{W}_t^{old}}; \mathcal{D}_t). \tag{2}$$

Then we set $\overline{W}_t = (\overline{W}_t^{old}, \overline{W}_t^{new})$. Finally, this candidate network $A_t$ with optimized weights $\overline{W}_t$ is tested on the validation set $\mathcal{V}_t$. The corresponding accuracy and network complexity is used to compute a reward $R$ (described in Section 3.2). The controller updates its policy based on $R$ and generates a new candidate network $A_t'$ to repeat the above procedure. After enough such interactions, the candidate architecture that achieves the maximal reward is the optimal one for task $t$, i.e. $A_t = (N_t, \overline{W}_t)$, where $N_t$ finally denotes the neurons of the optimal architecture.

**Controller Network**  The goal of the controller is to provide a neuron-level control that can decide which old neurons from previous tasks can be reused, and how many new neurons should be added. In our actual implementation, we assume there is a large hyper-network for the controller to search for a task network. Suppose the hyper-network has $l$ layers and each layer $i$ has a maximum of $u_i$ neurons. Each neuron has two actions, either "drop" or "use" (more actions for CNN, to be described later). Thus, the search space for the controller would be $2^n$ where $n = \sum_{i=1}^{l} u_i$ is the total number of neurons. Apparently, it is infeasible to enumerate all the action combinations and determine the best one. To deal with this issue, we treat the action sequence as a fixed-length string $a_{1:n} = a_1, a_2, \ldots, a_n$ that describes a task network. We design the controller as an LSTM network where each cell controls one $a_i$ in the hyper-network. Formally, we denote by $\pi(a_{1:n}|s_{1:n}; \theta_c)$ the policy function of the controller network as

$$\pi(a_{1:n}|s_{1:n}; \theta_c) = P(a_{1:n}|s_{1:n}; \theta_c) = \prod_{i=1}^{n} P(a_i|s_{1:i}; \theta_c). \tag{3}$$

The state $s_{1:n}$ is a sequence that represents *one* state; the output is the probability of a task network described by $a_{1:n}$; and $\theta_c$ denotes the parameters of the controller network. At this point we note that our model is a departure from standard models where states are considered individual $s_j$ and an episode is comprised of $s_{1:n}$. In our case we define $s_{1:n}$ as one state and episodes are created by starting with different initial states (described below).

Recall that in Fig.1, the two components in CLEAS work with each other *iteratively* and there are $\mathcal{H} \cdot \mathcal{U}$ such iterations where $\mathcal{H}$ is the number of episodes created and $\mathcal{U}$ the length of each episode.

Consider an episode $e = (s_{1:n}^1, \bar{a}_{1:n}^1, R^1, s_{1:n}^2, \bar{a}_{1:n}^2, R^2, \ldots, s_{1:n}^{\mathcal{U}}, \bar{a}_{1:n}^{\mathcal{U}}, R^{\mathcal{U}}, s_{1:n}^{\mathcal{U}+1})$. The initial state $s_{1:n}^1$ is either generated randomly or copied from the terminal state $s_{1:n}^{\mathcal{U}+1}$ of the previous episode. The controller starts with some initial $\theta_c$. For any $u = 1, 2, \ldots, \mathcal{U}$, the controller generates the most probable task network specified by $\bar{a}_{1:n}^u$ from $s_{1:n}^u$ by following LSTM. To this end, we use the recursion $a_j^u = f(s_j^u, h_{j-1}^u)$ where $h_{j-1}^u$ is the hidden vector and $f$ standard LSTM equations to generate $a_{1:n}^u$ from $s_{1:n}^u$. Let us point out that our RNN application $a_j^u = f(s_j^u, h_{j-1}^u)$ differs from the standard practice that uses $a_j^u = f(a_{j-1}^u, h_{j-1}^u)$. Action $\bar{a}_j^u$ is obtained from $a_{1:n}^u$ by selecting the maximum probability value for each $j$, $1 \leq j \leq n$. The task trains this task network, evaluates it on the validation set and returns reward $R^u$. We then construct $s_{1:n}^{u+1}$ from the previous neuron action $\bar{a}_j^u$ together with the layer index as $b_j^{u+1}$ for each $1 \leq j \leq n$. More concretely, $s_j^{u+1} = \bar{a}_j^u \oplus b_j^u$ where $\bar{a}_j^u, b_j^u$ have been one-hot encoded, and $\oplus$ is the concatenation operator. Finally, a new network architecture $\bar{a}_{1:n}^{u+1}$ is generated from $s_{1:n}^{u+1}$. At the end of each episode, the controller updates its parameter $\theta_c$ by a policy gradient algorithm. After all $\mathcal{H} \cdot \mathcal{U}$ total iterations, the task network that achieves the maximum reward is used for that task.

The choice for treating the state as $s_{1:n}$ and not $s_j$ has the following two motivations. In standard NAS type models after updating $s_j$ the network is retrained. This is intractable in our case as the number of neurons $n$ is typically large. For this reason we want to train only once per $s_{1:n}$. The second reason is related and stems from the fact that the reward is given only at the level of $s_{1:n}$. For this reason it makes sense to have $s_{1:n}$ as the state. This selection also leads to computational improvements as is attested later in Section 4.

**CLEAS-C for CNN** The design of CLEAS also works for CNN with *fixed* filter sizes where one filter corresponds to one neuron. However, we know that filter sizes in a CNN can have a huge impact on its classification accuracy. Therefore, we further improve CLEAS so that it can decide the best filter sizes for each task. In particular, we allow a new task to increase the filter size by *one* upon the previous task. For example, a filter size $3 \times 3$ used in some convolutional layer in task $t-1$ can become $4 \times 4$ in the same layer in task $t$. Note that for one task all the filters in the same layer must use the same filter size, but different layers can use different filter sizes.

We name the new framework as CLEAS-C. There are two major modifications to CLEAS-C. First, the output actions in the controller are now encoded by 4 bits and their meanings are "only use,""use & extend,""only drop" and "drop & extend" (see Fig. 2). Note that the extend decision is made at the neuron layer, but there has to be only one decision at the layer level. To this end, we apply simple majority voting of all neurons at a layer to get the layer level decision. The other modification regards the training procedure of the task network. The only different case we should deal with is how to optimize a filter (e.g. $4 \times 4$) that is extended from a previous smaller filter (e.g. $3 \times 3$). Our solution is to preserve the optimized parameters that are associated with the original smaller filter (the 9 weights) and to only optimize the additional weights (the $16 - 9 = 7$ weights). The preserved weights are placed in the center of the larger filter, and the additional weights are initialized as the averages of their surrounding neighbor weights.

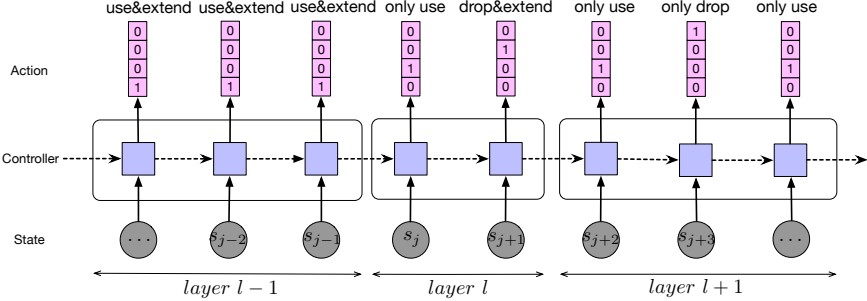

Figure 2: The controller design for convolutional networks.

## 3.2 TRAINING WITH REINFORCE

Lastly, we present the training procedure for the controller network. Note that each task $t$ has an independent training process so we drop subscript $t$ here. Within an episode, each action string $a_{1:n}^u$ represents a task architecture and after training gets a validation accuracy $\mathcal{A}^u$. In addition to accuracy,

we also penalize the expansion of the task network in the reward function, leading to the final reward

$$R^u = R(a_{1:n}^u) = \mathcal{A}(a_{1:n}^u) - \alpha\mathcal{C}(a_{1:n}^u) \tag{4}$$

where $\mathcal{C}$ is the number of newly added neurons, and $\alpha$ is a trade-off hyperparameter. With such episodes we train

$$J(\theta_c) = \mathbb{E}_{a_{1:n}\sim p(\cdot|s_{1:n};\theta_c)}[R] \tag{5}$$

by using REINFORCE. We use an exponential moving average of the previous architecture accuracies as the baseline.

We summarize the key steps of CLEAS in Algorithm 1 where $\mathcal{H}$ is the number of iterations, $\mathcal{U}$ is the length of episodes, and $p$ is the exploration probability. We point out that we do not strictly follow the usual $\epsilon$-greedy strategy; an exploration step consists of starting an epoch from a completely random state as opposed to perturbing an existing action.

---

**Algorithm 1:** CLEAS.

---

**Input:** A sequence of tasks with training sets $\{\mathcal{D}_1, \mathcal{D}_2, ..., \mathcal{D}_T\}$, validation sets $\{\mathcal{V}_1, \mathcal{V}_2, ..., \mathcal{V}_T\}$
**Output:** Optimized architecture and weights for each task: $A_t = (N_t, \overline{W}_t)$ for $t = 1, 2, \ldots, T$

**for** $t = 1, 2, \ldots, T$ **do**
    **if** $t = 1$ **then**
        Train the initial network $A_1$ on $\mathcal{D}_1$ with the weights optimized as $\overline{W}_1$;
    **else**
        Generate initial controller parameters $\theta_c$;
        **for** $h = 1, 2, \ldots, \mathcal{H}$ **do**
            /* A new episode */
            $w \sim \text{Bernoulli}(p)$;
            **if** $w = 1$ *or* $h = 1$ **then**
                /* Exploration */
                Generate a random state string $s_{1:n}^1$ but keep layer encodings fixed;
            **else**
                Set initial state string $s_{1:n}^1 = s_{1:n}^{\mathcal{U}+1}$, i.e. the last state of previous episode $(h-1)$;
            **for** $u = 1, 2, \ldots, \mathcal{U}$ **do**
                Generate the most probable action string $\bar{a}_{1:n}^u$ from $s_{1:n}^u$ by the controller;
                Configure the task network as $A^u$ based on $\bar{a}_{1:n}^u$ and train weights $W^u$ on $\mathcal{D}_t$;
                Evaluate $A^u$ with trained $\overline{W}^u$ on $\mathcal{V}_t$ and compute reward $R^u$;
                Construct $s_{1:n}^{u+1}$ from $\bar{a}_{1:n}^u$ and $b_{1:n}^u$ where $b_{1:n}^u$ is the layer encoding;
            Update $\theta_c$ by REINFORCE using $(s_{1:n}^1, \bar{a}_{1:n}^1, R^1, \ldots, s_{1:n}^{\mathcal{U}}, \bar{a}_{1:n}^{\mathcal{U}}, R^{\mathcal{U}}, s_{1:n}^{\mathcal{U}+1})$;
            Store $A^h = (N^{\bar{u}}, \overline{W}^{\bar{u}})$ where $\bar{u} = \arg\max_u R^u$ and $\bar{R}^h = \max_u R^u$;
        Store $A_t = A^{\bar{h}}$ where $\bar{h} = \arg\max_h \bar{R}^h$;

---

## 4 EXPERIMENTS

We evaluate CLEAS and other state-of-the-art continual learning methods on MNIST and CIFAR-100 datasets. The key results delivered are model accuracies, network complexity and training time. All methods are implemented in Tensorflow and ran on a GTX1080Ti GPU unit.

### 4.1 DATASETS AND BENCHMARK ALGORITHMS

We use three benchmark datasets as follows. Each dataset is divided into $T = 10$ separate tasks. MNIST-associated tasks are trained by fully-connected neural networks and CIFAR-100 tasks are trained by CNNs.

**(a) MNIST Permutation** (Kirkpatrick et al., 2017): Ten variants of the MNIST data, where each task is transformed by a different (among tasks) and fixed (among images in the same task) permutation of pixels. **(b) Rotated MNIST** (Xu & Zhu, 2018): Another ten variants of MNIST, where each task is rotated by a different and fixed angle between 0 to 180 degree. **(c) Incremental CIFAR-100** (Rebuffi et al., 2017): The original CIFAR-100 dataset contains 60,000 32×32 colored images that belong to 100 classes. We divide them into 10 tasks and each task contains 10 different classes and their data.

We select four other continual learning methods to compare. One method (**MWC**) uses a fixed network architecture while the other three use expandable networks.

**(1) MWC:** An extension of EWC (Kirkpatrick et al., 2017). By assuming some extent of correlation between consecutive tasks it uses regularization terms to prevent large deviation of the network weights when re-optimized. **(2) PGN:** Expands the task network by adding a fixed number of neurons and layers (Rusu et al., 2016). **(3) DEN:** Dynamically decides the number of new neurons by performing selective retraining and network split (Yoon et al., 2018). **(4) RCL:** Uses NAS to decide the number of new neurons. It also completely eliminates the forgetting problem by holding the previous neurons and their weights unchanged (Xu & Zhu, 2018).

For the two MNIST datasets, we follow (Xu & Zhu, 2018) to use a three-layer fully-connected network. We start with 784-312-128-10 neurons with RELU activation for the first task. For CIFAR-100, we develop a modified version of LeNet (LeCun et al., 1998) that has three convolutional layers and three fully-connected layers. We start with 16 filters in each layer with sizes of $3 \times 3$, $3 \times 3$ and $4 \times 4$ and stride of 1 per layer. Besides, to fairly compare the network choice with (Xu & Zhu, 2018; Yoon et al., 2018), we set: $u_i = 1000$ for MNIST and $u_i = 128$ for CIFAR-100. We also use $\mathcal{H} = 200$ and $\mathcal{U} = 1$. The exploration probability $p$ is set to be 30%. We select the RMSProp optimizer for REINFORCE and Adam for the training task.

We also implement a version with states corresponding to individual neurons where the controller is following $a_j^u = f(a_{j-1}^u, h_{j-1}^u)$. We configure this version under the same experimental settings as of CLEAS and test it on the three datasets. The results show that compared to CLEAS, this version exhibits an inferior performance of **-0.31%**, **-0.29%**, **-0.75%** in relative accuracy, on the three datasets, respectively. Details can be found in Appendix.

## 4.2 EXPERIMENTAL RESULTS

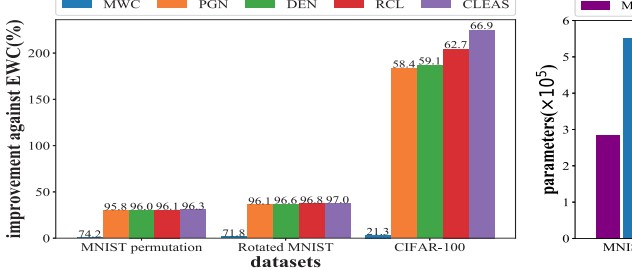
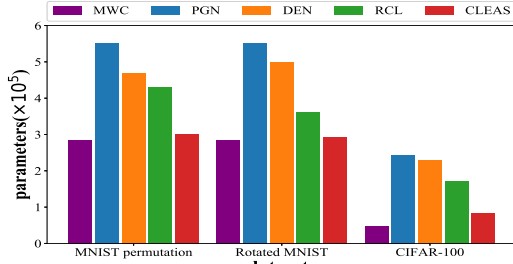

Figure 3: Average test accuracy across all tasks.    Figure 4: Average number of parameters.

**Model Accuracy**    We first report the averaged model accuracies among all tasks. Fig.3 shows the relative improvements of the network-expandable methods against EWC (numbers on the top are their absolute accuracies). We clearly observe that methods with expandability can achieve much better performance than MWC. Furthermore, we see that CLEAS outperforms other methods. The average relative accuracy improvement of CLEAS vs RCL (the state-of-the-art method and the second best performer) is **0.21%**, **0.21%** and **6.70%**, respectively. There are two reasons: (1) we completely overcome the forgetting problem by *not* altering the old neurons/filters; (2) our neuron-level control can precisely pick useful old neurons as well as new neurons to better model each new task.

**Network Complexity**    Besides model performance, we also care about how complex the network is when used to model each task. We thus report the average number of model weights across all tasks in Fig. 4. First, no surprise to see that MWC consumes the least number of weights since its network is non-expandable. But this also limits its model performance. Second, among the other four methods that expand networks we observe CLEAS using the least number of weights. The average relative complexity improvement of CLEAS vs RCL is **29.9%**, **19.0%** and **51.0%** reduction, respectively. It supports the fact that our NAS using neuron-level control can find a very efficient architecture to model every new task.

**Network Descriptions**    We visualize some examples of network architectures the controller generates. Fig. 5 illustrates four optimal configurations (tasks 2 to 5) of the CNN used to model CIFAR-100. Each task uses three convolutional layers and each square represents a filter. A white square means it is not used by the current task; a red square represents it was trained by some earlier task and now reused by the current task; a light yellow square means it was trained before but not reused; and a

dark yellow square depicts a new filter added. According to the figure, we note that CLEAS tends to maintain a concise architecture all the time. As the task index increases it drops more old filters and only reuses a small portion of them that are useful for current task training, and it is adding fewer new neurons.

**CLEAS-C**   We also test CLEAS-C which decides the best filter sizes for CNNs. In the CIFAR-100 experiment, CLEAS uses fixed filter sizes $3 \times 3$, $3 \times 3$ and $4 \times 4$ in its three convolutional layers. By comparison, CLEAS-C starts with the same sizes but allows each task to increase the sizes by one. The results show that after training the 10 tasks with CLEAS-C the final sizes become $4 \times 4$, $8 \times 8$, and $8 \times 8$. It achieves a much higher accuracy of **67.4%** than CLEAS (**66.9%**), i.e. a **0.7%** improvement. It suggests that customized filter sizes can better promote model performances. On the other hand, complexity of CLEAS-C increases by 92.6%.

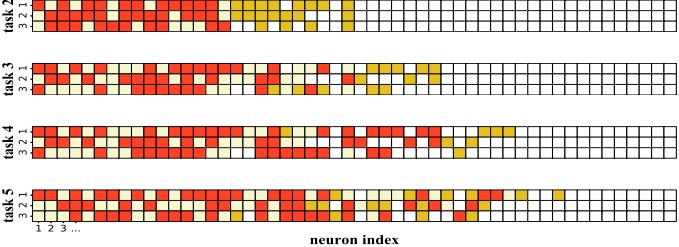

Figure 5: Examples of CNN architectures for CIFAR-100.

**Neuron Allocation**   We compare CLEAS to RCL on neuron reuse and neuron allocation. Fig. 6 visualizes the number of reused neurons (yellow and orange for RCL; pink and red for CLEAS) and new neurons (dark blue for both methods). There are two observations. On one hand, CLEAS successfully drops many old neurons that are redundant or useless, ending up maintaining a much simpler network. On the other hand, we observe that both of the methods recommend a similar number of new neurons for each task. Therefore, the superiority of CLEAS against RCL lies more on its selection of old neurons. RCL blindly reuses all previous neurons.

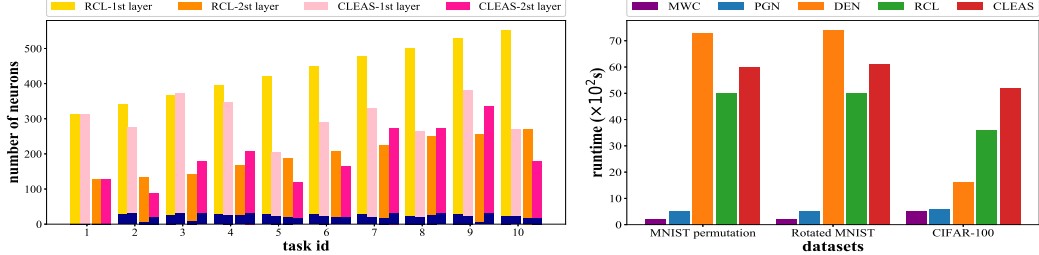

Figure 6: Neuron allocation for MNIST Permulation.          Figure 7: Training time

**Training Time**   We also report the training time in Fig.7. It is as expected that CLEAS' running time is on the high end due to the neuron-level control that results in using a much longer RNN for the controller. On the positive note, the increase in the running time is not substantial.

**Hyperparameter Sensitivity**   We show the hyperparameter analysis in Appendix. The observation is that the hyperparameters used in CLEAS are not as sensitive as those of DEN and RCL. Under all hyperparameter settings CLEAS performs the best.

## 5   CONCLUSIONS

We have proposed and developed a novel approach CLEAS to tackle continual learning problems. CLEAS is a network-expandable approach that uses NAS to dynamically determine the optimal architecture for each task. NAS is able to provide a neuron-level control that decides the reusing of old neurons and the number of new neurons needed. Such a fine-grained control can maintain a very concise network through all tasks. Also, we completely eliminate the catastrophic forgetting problem by never altering the old neurons and their trained weights. With demonstration by means of the experimental results, we note that CLEAS can indeed use simpler networks to achieve yet higher model performances compared to other alternatives. In the future, we plan to design a more efficient search strategy or architecture for the controller such that it can reduce the runtime while not compromising the model performance or network complexity.

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
