# OpenReview forum: "Efficient Architecture Search for Continual Learning"
_ICLR.cc/2021/Conference — Reject_

### Official Review · AnonReviewer3 · 2020-10-25
**Extension of reinforced continual learning (RCL)**

**Rating:** 6
**Confidence:** 3

**Review:**

This paper falls into a class of continual learning methods which accommodate for new tasks by expanding the network architecture, while freezing existing weights. This freezing trivially resolves forgetting. The (hard) problem of determining how to expand the network is tackled with reinforcement learning, largely building upon a previous approach (reinforced continual learning, RCL). Apart from some RL-related implementation choices that differ here, the main difference to RCL is that the present method learns a mask which determines which neurons to reuse, while RCL only uses RL to determine how many neurons to add. Experiments demonstrate that this allows reducing network size while significantly improving accuracy on Split CIFAR-100. The runtime is, however, increased here.

The obvious downside to this approach (and to RCL) is the potentially very increase in runtime stemming from RL, which requires fully training many networks just to solve one additional task. This renders the approach impractical for large models; consistent with this, the authors only study models of modest dimensions.

The present paper is mostly an extension of RCL. Thus, limited novelty is its main weakness. But the experimental gains are significant, the extension to RCL is meaningful and the paper is easy to follow.

I leave some questions and comments for the authors below:
- Did the authors re-implement the RCL objective using their own RL algorithmic choices? The RL implementation in the RCL paper differs in many ways (for example, actor-critic learning is used), which leaves some doubt as to whether (part of) the benefits stem from the changes to RL, or from the actual neuron-level control proposed here. I would like to hear the authors' reply to this point.

- "We point out that we do not strictly follow the usual $\epsilon$-greedy strategy; an exploration step consists of starting an epoch from a completely random state as opposed to perturbing an existing action."
Isn't this still $\epsilon$-greedy? I do have a question on this point, though: is the exploration probability annealed? Picking random states (with a high probability of 30%) seems very extreme. Can the authors provide learning curves for the controller?

- Permuted MNIST is admittedly not a great dataset to study transfer learning. Can the authors repeat the analysis of Figure 6 on Split CIFAR-100? Is allocation decreasing as new tasks are learned, suggesting that some form of transfer is occurring at the architecture search level, or does it remain roughly constant, on that dataset? Still on transfer learning: it would be good to report the actual training curve of the resulting nework (not the controller) over tasks and investigate whether learning becomes faster as more CIFAR splits are learned, compared to a baseline which does not benefit from architecture search. This would make the paper much stronger, in my opinion.

- MWC: how is this method different from standard EWC? I couldn't find any explanation.

- While EWC is a relevant baseline, it would be good to report as well the performance of simple joint multitask training as an upper baseline.

- "The results show that compared to CLEAS, this version exhibits an inferior performance of -0.31%, -0.29%, -0.75% in relative accuracy"
What is relative accuracy? Relative to MWC, as in Fig. 3? In any case, as these improvements are somewhat modest, how does runtime compare for the two options?

- Regarding training time, Fig. 7: while runtime in seconds is important, can the number of (controller network) training iterations of RCL vs. CLEAS be provided as well?

- Readability could be improved by using \citep{} instead of \citet{}.

--
Post-rebuttal edit: I read the authors' reply and thank them for the clarifications. I maintain my score of 6.

---

> ### Author Response · Authors · 2020-11-17
> **Detials about our CLEAS**
>
> Thank you very much for reviewing the paper.
>
> Yes, we have re-implemented the RCL algorithm following their released code for MNIST; we appreciate that the authors of RCL provided the source code for CIFAR dataset training. We found that RCL uses actor-critic learning to allocate the number of new neurons. RCL aims to control the allocation of new neurons, but it lacks control on old neurons. The action space in RCL is rather small, e.g., there are only 30 action candidates in their method. As such, RCL can use a conventional reinforcement learning method to define states and actions for policy optimization. Actually, the task network in the RCL model can always be expanded, whereby the new task training inherits a part of the benefits from the expanded network. However, CLEAS under specially designed reinforcement learning pays more attention to finding a better sub-architecture from both old and new neurons because of its neuron-level control policy. CLEAS aims to search a better architecture, which selects part of old and new neurons, for a given task from a very large search space. CLEAS needs to use a reinforcement learning method efficiently to search for a better neuron-control policy from the large search space and unobserved architecture candidates.
>
> Since the action in CLEAS depends on the previous state, we attempt to generate random actions with an initial probability (exploration probability) to avoid local minima; the exploration probability can be annealed automatically in CLEAS. The initial probability of 30% is an experimental setting from select trials.
>
> Yes, as Figure 5 shows, since we have shown the selected structure of CNN for CIFAR-100, thereby we only provide the result of Permuted MNIST in Figure 6 due to the page limit. In CLEAS, as new sequential tasks keep requiring the task network for training, each new task only needs a small part of old neurons after receiving a network description from the controller. Besides, it only needs some of the new neurons to obtain better test accuracy even if it has many new neurons at disposal.
>
> EWC is a standard continual learning algorithm, which only enforces the parameter learning in the new upcoming task to be close to the current task’s parameters. While MWC is an extended version of EWC, which also considers the inherent relationships between the current task and the new upcoming task. In the implementation, MWC adds regularization terms to learn such correlations among tasks. We will update this point in Section 4.1 to make it clearer.
>
> We will add the EWC algorithm as an additional baseline. Thank you very much for this suggestion.
>
> We implemented a version with states corresponding to individual neurons, where it considers each output of an RNN-based network as an individual action and it starts with only one initial state s0. We define s1:n as one state in CLEAS. Hence, the relative accuracy denotes the results based on the former version. We also show the differences between these two models in the appendix. The experimental details are also provided in the appendix.
>
> We set the same number of training epochs in RCL and CLEAS. Therefore, we record the runtime for baselines and CLEAS to make a fair comparison.
>
> We use the \citep{} in the edited version of the paper.
>
> Thank you very much for your suggestions and articulated questions and comments.

---

### Official Review · AnonReviewer4 · 2020-10-27
**A method for continual learning in feedforward neural networks that requires only moderate network growth.**

**Rating:** 6
**Confidence:** 4

**Review:**

Summary: This paper presents a new method for continual learning by examining for each new task the NN created so far, deciding for each neuron whether to keep it (with the same weights), and how many new neurons should be added. This is done by a controller networks, consisting of LSTM units.

Pros: They demonstrate improved performance in comparison with 3 previously proposed methods, in particular MWC, an improvement (invented and tested by whom?)  over EWC of (Kirkpatrick et al., PNAS 2017), while using about the same number of parameters.

Cons: Their method is quite complicated and somewhat opaque to me, and it requires substantially more compute, see Fig. 7 (although it is stated on p. 8 that the „increase in running time is not substantial“; I do not understand that). Also the controller network needs to be apparently be trained for the whole task sequence. If this is correct, the full task sequence has to be known in advance.

I do not understand the statement in the Discussion „we completely eliminate the catastrophic forgetting problem by never altering the old neurons and their trained weights“. If one removes some neurons in a network (only feedforward networks are considered), one changes the input to downstream neurons.

---

> ### Author Response · Authors · 2020-11-17
> **The concerns about baselines and catastrophic forgetting problem**
>
> Thank you very much for carefully reviewing the paper. MWC is an extended version of EWC, which also considers the inherent correlations between the current task and the upcoming task besides enforcing the parameter learned in the upcoming task to be close to the current task’s parameters. (Zhang Jie et al., Regularize, expand and compress: Non expansive continual learning. We are much obliged to the authors for providing the source code). Although CLEAS’ running time is higher than the RCL’s time due to the neuron-level control that results in using a longer RNN for the controller, CLEAS aims to discover the optimal sub-structure for the task network while previous works only reuse all of the trained neurons for the new upcoming task. Hence, CLEAS can cope with the forgetting problem. Besides, CLEAS provides a simpler controller network with far fewer parameters than RCL. For example, as Figure 1 and Figure 2 show, our output layer in the controller has only two or four digits. Therefore, the increase in running time is not substantial.
>
> The controller like NAS does not need to know the whole task sequence in advance; our controller is based on a hyper-network that is defined based on the available computing resources. CLEAS controls the new neuron allocation by reinforcement learning to decrease the waste of computing resources. Meanwhile, CLEAS avoids the forgetting issue related to the old tasks.

---

### Official Review · AnonReviewer2 · 2020-10-27
**Correct paper that still need some work**

**Rating:** 4
**Confidence:** 5

**Review:**

The authors tackle the problem of efficient architecture search for continual learning. They propose to use reinforcement learning-based neural architecture search to efficiently expend layers, select which neurons trained on previous tasks to reuse and which new neurons to train from scratch. They validate the approach using standard experiments on MNIST and Cifar-100. The paper is well written, clearly explaining each component and the reasons for their choices.

My main concern is about efficiency, the choice of RNN generating architecture propositions and using the performance of each proposition as a reward can't really be called efficient as it requires multiple (200 in the article) full trainingC on each task.  It would be interesting to see a training time comparison of the proposed method against the other baselines (e.g. measuring the training time as (i)wall-clock time or (ii)total number of parameter updates including the NAS part of the training).

Questions:
- If my understanding is correct, the RNN is trained on the validation set (section 3.1). What is the size of this validation set for each task and what is the protocol to prevent overfitting? Are the HP tuned on the same set?
- For the controller network training, why using an additional parameter for the exploration instead of directly following the policy parametrized by $\theta_c$ ?
- Why using LeNet architecture as the backbone architecture instead of a more recent and commonly used model in CL (e.g. smaller Resnets-18 or 34)?

I would also like to see how a simple baseline like random search perform, using the same number of models as the NAS approach (i.e. training 200 networks sampled from the search space used by the NAS procedure on each task).

---

> ### Author Response · Authors · 2020-11-17
> **concerns about efficiency**
>
> Thanks very much for taking the time to review our paper. The main contribution of CLEAS is to provide a more accurate neural network search method. It aims at discovering an optimal sub-structure from the trained neurons, and meanwhile, it can make a reinforced choice by combining additional new neurons for new task knowledge learning. The RNN-based controller in our CLEAS brings a tiny runtime increase compared to RCL’s RNN controller since the length of our RNN is longer than RCL. But our controller is simpler than RCL. Our controller uses fewer parameters. For example, the output size of the controller for MNIST is two digits while the output size in RCL is much larger than ours. Therefore, we observe that the increase in the running time is not substantial after we compute the training time of all methods.
>
> (1) Yes, we evaluate the quality of our designed task model by the validation set. Each task in MNIST Permutations or MNIST Rotations contains 55,000 training samples, 5,000 validation samples. and 10,000 test samples. Each task in CIFAR-100 contains 5,000 samples for training and 1,000 for testing. We randomly select 1,000 samples from each task training sample as the validation samples and assure each class in a task has at least 100 validation samples. The model observes the tasks one by one and does not see any data from previous tasks. We put this explanation in our Appendix part. We consider each task is an individual one, thereby each task has its own validation set and testing test. To avoid the overfitting problem, in task network training, we gradually and automatically decrease the learning rate.
>
> (2)In our CLEAS, every action depends on the previous state, for more effective training, we generate random actions with a defined probability to avoid local minimums.
>
> (3)To fairly compare with previous works, e.g., RCL, DEN, we use the similar LeNet as our CNN-based task network in which the filter number and the filter size can be controlled by our CLEAS-C.
>
> Thanks for the great suggestion. We will provide a simple baseline like a random search in our new version.

---

> > ### Comment · AnonReviewer2 · 2020-11-23
> > **Any update on the revision ?**
> >
> > Thank you for your answers.
> > Since the title mentions efficiency, I really think that this should be quantitatively addressed in the paper or at least discussed. Not only by comparing the proposed approach to RCL which is itself very inefficient (training 50 models per task on MNIST !) but also to other methods from the continual learning literature.
> >
> > (3) I agree that this fair comparison is necessary but I think that showing the ability of the proposed approach to scale to more modern architecture would be a large improvement for the paper since VGG and Resnet architectures are mentioned in RCL but the experiments are also limited to LeNet.
> >
> > - Do you have any update on the random search baseline?

---

> > > ### Author Response · Authors · 2020-11-24
> > > **Further details about CLEAS**
> > >
> > > Thanks for the kind reply. We really appreciate the significant and constructive suggestions. We consider the efficiency in this paper is to find a better architecture for task training and obtain a better test accuracy. Hence, designing a promising framework based on your great suggestion to easily and quickly get a good architecture for task network will inspire our future work. The 'efficiency' in your understanding is more accurate. We will more focus on the efficiency of continual learning. We have uploaded the new version of our paper, which aims to address the concerns of all the reviewers. Since the limits of the pages, we put the simple baseline in 'CLEAS vs. Standard NAS Controller' of Appendix. Since we also implement CLEAS with a standard NAS and show the details about this model. To this end, we put the random search method here to make further discussion could be appropriate.

---

### Official Review · AnonReviewer1 · 2020-10-29
**Solving continual learning problems with minimal expansion of network parameters.**

**Rating:** 6
**Confidence:** 3

**Review:**

Efficient Architecture Search for Continual Learning
- Summary
This paper aims to solve continual learning problems with minimal expansion of network parameters. The authors propose Continual Learning with Efficient Architecture Search (CLEAS), which is equipped with a neuron-level NAS controller. The controller selects 1) the most useful previous neurons to model the new task (knowledge transfer) and 2) a minimum number of additional neurons. The experimental results show that the proposed method outperforms state-of-the-art methods on several continual learning benchmark tasks such as MNIST Permutation, Rotation MNIST, and Incremental CIFAR-100.

- Strong points
	1. The proposed framework selectively comprises neurons for new tasks, and training only newly added weights enables zero-forgetting of previously learned tasks.
	2. The experimental results showed performance improvements compared with the previous algorithms (PGN, DEN, RCL) while preserving or reducing the number of parameters, especially in the case of CIFAR-100.

- Weakness
	1. From the perspective of real-world problems, the neuron-level decision through the RNN controller costs a long training time. The authors in this paper demonstrated small-scale neural networks such as 3-layers. Although the authors mentioned like “On the positive note, the increase in the running time is not substantial.”, I’m wondering it is plausible, and I couldn’t find what the running time in the figure 7 means.
	2. It is not clear for me the rationale behind the sequential states of neurons and the authors’ claim that “This state definition deviates from the current practice in related problems that would define a state as an observation of a single neuron.”. Also, what does “standard model” mean in page 4? Is the sequential state invariant to permutations in neuron topology?

- Questions
	1. How did you set the maximum of u_i neurons in your experiments?
	2. Do you have any plan to publish the source codes for reproducibility?
	3. Please address and clarify the cons above.

- Additional feedback
	- Typo: wrong citation for ENAS (line 7, “Neural Architecture Search” paragraph in section 2)
	- I would like to recommend denoting the accuracy of MWC in section 4.2.

---

> ### Author Response · Authors · 2020-11-17
> **Details about our CLEAS**
>
> Thank you very much for taking the time to review our paper. We use an RNN in the paper as our controller network where each output determines the probability of the corresponding task network neuron being selected. Compare to the best baseline RCL, the length of our RNN is longer than RCL (longer input sequences). However, our controller is simpler than the one used by RCL since it is using drastically fewer parameters. For example, the output size of the controller for MNIST consists of two digits while the output size in RCL is much larger than ours. Therefore, we observe that the increase in the running time is not substantial. Besides, as Figure 3 and Figure 4 show, CLEAS selects a much simpler task network architecture for task training and nevertheless obtains better test results.
>
> In CLEAS, each state corresponds to a task network. In standard controllers (e.g., RCL), the states correspond to individual sj (i.e.., each state denotes a single neuron) and an episode is comprised of s1:n. This is intractable in our case as the number of neurons is typically large. For this reason we want to train only once per s1:n. The second reason is related and stems from the fact that the reward is given only at the level of s1:n. We also use a standard model as our controller in CLEAS, and provide the experimental results in Sec. 4.1. The details are presented in the Appendix due to the limit on the number of pages.
> -Regarding the u_i setting, we set our controller as a fixed hyper-network considering the scenario that we only have limited computing resources. Hence, we set: u_i = 1000 for MNIST and u_i = 128 for CIFAR-100.
>
> -Yes, we will release the source codes in github after the paper is accepted.
>
> Thank you very much for your kind feedback; we have revised the paper to account for the typos.

---

### Decision · Program_Chairs · 2021-01-07
**Final Decision**

**Decision:**

Reject

**Comment:**

The authors propose a network-expandable approach to tackle NAS in the continual learning setting. More specifically, they use a RNN controller to decide which neurons to use (for a new task) and the additional capacity required (i.e., number of new neurons to add). This work can be viewed as an extension of RCL and as such suffers from the large runtime. This was a concern for most reviewers. While reviewers highlighted the gains in the experiments conducted, several questions remained regarding the efficiency of the proposed approach and how it compares to other strategies. The practical relevance of the proposed approach was also a concern as its application requires to restrict it to models of modest size.